# The influence of magnocellular and parvocellular visual information on global processing in White and Asian populations

**Tiffany A. Carther-Krone**⬥\*, **Jonathan J. Marotta**

Perception and Action Lab, Department of Psychology, University of Manitoba, Winnipeg, Manitoba, Canada

\* lazart@myumanitoba.ca

**Data Availability Statement:** All data will be available via dataverse (https://doi.org/10.34990/FK2/WTPDB5).

**Funding:** This research was supported by a grant from the Natural Science and Engineering

## Abstract

Humans have the remarkable ability to efficiently group elements of a scene together to form a global whole. However, cross-cultural comparisons show that East Asian individuals process scenes more globally than White individuals. This experiment presents new insights into global processing, revealing the relative contributions of two types of visual cells in mediating global and local visual processing in these two groups. Participants completed the Navon hierarchical letters task under divided-attention conditions, indicating whether a target letter "H" was present in the stimuli. Stimuli were either 'unbiased', displayed as black letters on a grey screen, or biased to predominantly process low spatial frequency information using psychophysical thresholds that converted unbiased stimuli into achromatic magnocellular-biased stimuli and red-green isoluminant parvocellular-biased stimuli. White participants processed stimuli more globally than Asian participants when low spatial frequency information was conveyed via the parvocellular pathway, while Asian participants showed a global processing advantage when low spatial frequency information was conveyed via the magnocellular pathway, and to a lesser extent through the parvocellular pathway. These findings suggest that the means by which a global processing bias is achieved depends on the subcortical pathway through which visual information is transmitted, and provides a deeper understanding of the relationship between global/local processing, subcortical pathways and spatial frequencies.

## Introduction

Every day the human visual system is bombarded by vast amounts of visual information. Typically developing individuals are able to quickly and efficiently group individual features of a scene together to form a global whole, a phenomenon known as the 'global precedence effect' [1–3]. However, recent research has shown that the relative preference for global versus local distribution of attention may differ based on culture.

Comparisons between White (i.e., Canadians, Americans, British and Australians) and East Asian populations (i.e., Chinese and Japanese) show that East Asian individuals process objects and scenes more globally than White individuals ([4–7] although see [8–12]). For example,

Research Council of Canada (NSERC) (Grant no. 04964-2018) held by J.J.M. The funders had no role in study design, data collection and analysis, decision to publish, or preparation of the manuscript.

**Competing interests:** The authors have declared that no competing interests exist.

when individuals are asked to make absolute judgments (requiring local processing) and relative judgments (requiring global processing), East Asian individuals perform better than White individuals at the relative task while White individuals perform better than East Asian individuals at the absolute task [13]. Similarly, an fMRI study showed that achieving equivalent levels of behavioral performance required more sustained attentional effort for the absolute task in East Asian individuals and for the relative task in White individuals [14]. When the global processing bias is directly compared between White and East Asian individuals, a global advantage is found in East Asian relative to White individuals, extending through to a second generation of Asian-Australians [15]. An electrophysiological study examining the neural mechanisms and the temporal dynamics related to a global processing bias has shown a greater sensitivity to global congruency in East Asian relative to White individuals, as indexed by an early P1 component [16].

Studies involving eye movements have suggested that perceptual differences may be driven by culture-specific tuning towards visual spatial frequency information [17]. Face recognition studies show that White individuals preferentially process high spatial frequency information from foveal vision and are more biased to local processing in hierarchical stimuli, while East Asian individuals preferentially process contextual information by relying on extra-foveal vision during face recognition, favoring globally-based holistic stimulus processing [17]. East Asian individuals also show more of a reliance on extra-foveal vision than White individuals when detecting low level visual stimuli [18] and for change detection of complex real-world stimuli [5], indicating that East Asian individuals allocate their attention more broadly than White individuals.

Numerous psychophysical studies have shown that the global precedence effect in typically developing individuals is mediated by low spatial frequencies [19–21]. Studies have also shown that stimuli presented without low spatial frequency information do not result in a global precedence effect [22, 23], but rather result in a local precedence effect [24]. Since White and East Asian individuals both show a global precedence effect, it is unlikely that differences in global processing are due to an inability to process low spatial frequency information. Based on previous research showing that culture-specific tuning towards visual spatial frequency information can influence perception, we explore the notion that differences in global processing are driven by differences in the means by which low spatial frequency information is conveyed through the visual system.

Before visual information makes its way from visual cortex to extrastriate visual areas by the dorsal stream, which governs the visual control of action, and the ventral stream, which governs visual perception, it initially passes from the retina to the primary visual cortex via the relay station called the lateral geniculate nucleus [25, 26]. The lateral geniculate nucleus consists of two pathways, magnocellular and parvocellular, that operate relatively independently in early visual processing and encode contrasting information. Specifically, magnocellular pathway neurons are visual cells that are not responsive to color and are known for processing achromatic, low contrast stimuli [27, 28], which is critical for global processing [20, 24]. In contrast, parvocellular pathway neurons are visual cells that are highly responsive to opposing colours (red-green or blue-yellow) [29] and high spatial frequencies [30, 31], requiring much higher contrast (~8% at least) when detecting achromatic stimuli [32]. While both magnocellular and parvocellular cells are present in both dorsal and ventral processing streams [27, 28], magnocellular cells are primarily conveyed through the dorsal stream, responsible for global processing, and parvocellular cells are primarily conveyed through the ventral stream, responsible for local processing.

A powerful way to gain insight into the way global information is processed in the visual system is to examine how visual information is processed through the magnocellular and

parvocellular streams. For example, when stimuli are biased so that they are processed primarily through either the magnocellular or parvocellular pathway, global processing abilities differ between the two pathways in both individuals with simultanagnosia [33] and autism spectrum disorder [34]. This biasing can be achieved by manipulating the spatial frequencies conveyed through the parvocellular pathway. Although an increased sensitivity for high spatial frequencies is crucial for local processing via the parvocellular pathway specifically, the parvocellular pathway's sensitivity to color information can also be selectively stimulated to act as a low-pass filter, conveying low spatial frequency information (required for global processing) when stimulated with isoluminant color stimuli [19, 35]. Although the magnocellular pathway is considered the primary pathway through which global information is transmitted, biasing both pathways to convey global information provides new insights into the relative contribution of these cell types to visual perception. Here we investigate the relative contribution of these two types of visual cells to the global precedence effect across Asian and White groups.

To examine the contribution of magnocellular and parvocellular cells to the global precedence effect, we used the Navon letters task [3] to compare global/local processing differences between White and Asian groups. Participants completed the Navon task under divided-attention conditions, indicating whether a target letter "H" was present in the hierarchical stimuli. Based on previous research [15] we expected that overall, Asian participants would show a stronger global advantage than White participants as indexed by faster reaction times, lower number of errors made, and lower inverse efficiency scores when the target was presented at a global compared to a local level. We also used psychophysical techniques to examine the mechanism underlying differences in global/local processing between groups by biasing stimuli to test the indirect contribution of the dorsal and ventral pathways in mediating global and local visual processing. For each participant we first established the achromatic contrast threshold and chromatic isoluminance threshold [36]. Next, to test global and local visual processing, we used these thresholds to dynamically generate magnocellular- and parvocellular-biased stimuli from a set of 'unbiased' hierarchical letter stimuli. Considering the parvocellular system's ability to convey low spatial frequencies when stimulated with isoluminant color stimuli, both magnocellular and parvocellular systems were biased to activate in response to the same range of spatial frequencies to test the relative contributions of the dorsal and ventral pathways in mediating global visual processing in White compared to Asian participants. Since global processing relies on low spatial frequency information and previous research [17] has shown that White individuals preferentially process high spatial frequency information, which generally relies on the parvocellular system, we hypothesized that global processing in White participants would be more influenced by the parvocellular system than the magnocellular system when the ventral stream is biased to convey low spatial frequency information via the parvocellular pathway. By filtering out the high-spatial frequency information (required for local processing) from the parvocellular pathway, such that mainly low-frequency information (required for global processing) would pass through, we hypothesized that in the parvocellular-biased condition White participants would process the target faster and more accurately at the global level than the local level. Similarly, since Asian individuals have been shown to rely more on low contrast stimuli, which involves mainly the magnocellular pathway, we expected that in the magnocellular-biased condition Asian participants would process the target faster and more accurately at the global level than the local level. By filtering out high-spatial frequency information from the parvocellular stimuli, this allowed us to compare global processing abilities overall when stimuli are biased to one of the two subcortical pathways (magnocellular/parvocellular) as a way to examine if any differences in processing ability between the two subcortical pathways are responsible for the global processing advantage commonly found in the research.

## Method

### Participants

This study tested two groups: (a) 27 White participants (17 females) with a mean age of 23.5 years (18–36 years old) and (b) 25 Asian participants (15 females) with a mean age of 21.4 years (18–28 years old). The number of participants was determined based on comparable research examining cultural differences in global processing [15, 16] and magnocellular functioning [37]. Based on this resulting sample size, a sensitivity analysis for the key prediction (i.e., the group x condition x level interaction) was conducted. G*Power 3.1. [38] was used to conduct an F-test for repeated measures ANOVA, specifying a within-between interaction with 2 groups (Asian, White) and 6 measurements (2 levels x 3 conditions). Assuming an error probability of .05 and a nonsphericity correction of 1, the study was 80% powered to detect an effect size $\eta_p^2 = .021$.

All participants were right-handed young adults with normal or corrected-to-normal vision. Ethnicity was determined by self-report and all participants were Introduction to Psychology students studying at an English-language university. All White participants and 11 Asian participants indicated English as their first language, while 14 Asian participants indicated Chinese as their first language. However, based on the rigorous English language testing required before being accepted into a University of Manitoba degree program, all participants were expected to be highly fluent at identifying single English letters, although no direct information is available regarding each participant's specific English language ability. Participants were excluded from the study if they failed a colorblindness test administered before the start of the experiment (N = 7). Participants signed an informed consent form in writing before taking part in the study, which was approved by the Research Ethics Board (REB1) at the University of Manitoba.

### General procedure

Participants signed a consent form and completed a short demographics questionnaire. They then completed the experiment, which began with a color-blindness test, followed by a luminance contrast thresholding task of which the results were used to create the magnocellular-biased stimuli and an isoluminance task of which the results were used to create the parvocellular-biased stimuli. The final portion of the experiment involved a computerized task using the hierarchical letters task [3].

### Psychophysical thresholds

To determine the luminance contrast for each participant, a multiple staircase procedure was used to find the luminance threshold. Participants were shown light gray hierarchical stimuli overlaid on a dark gray background and asked if the stimuli was detected following each stimulus presentation. On 25% of the trials no stimulus was presented, which served as catch trials. Each stimulus was presented for 1500 ms, during which a response was made. If the stimuli was detected, the contrast between the stimuli and background was decreased in the next trial by modifying the luminance of the stimuli. Otherwise the contrast was increased. A commonly used luminance threshold-finding algorithm [36] was used to compute the mean of the turn-around points above and below medium-gray background. A luminance (~3.5% Weber contrast) value was then computed from this threshold for the grayscale stimuli and was used in the low-luminance-contrast (magnocellular-biased) condition.

For the chromatically defined stimuli in the isoluminant (parvocellular-biased) condition, the isoluminance point was found using heterochromatic flicker photometry [39, 40] with

stimuli consisting of the same hierarchical letters from the luminance contrast task displayed in alternating colors (pure red and green). By alternating the two colors in the range of 12–20 Hz, the flicker disappears for a small range of luminance values. The color values at the point where the two colors appear to fuse together and the stimulus appears steady is the participant's isoluminance interval. In this task participants used the up and down arrow keys to adjust the green color to the point at which the stimulus appeared steady. Depending on the response, the output of the green color was adjusted up or down such that the participant passed over the isoluminant point many times, and the average of the values in the narrow range when the participant reported a steady stimulus was computed as the isoluminance value for that participant. This luminance contrast threshold and isoluminance value was then used to dynamically create the magnocellular- and parvocellular-biased stimuli respectively for the experiment that followed.

## Stimuli and procedure

All experiments were programmed using the Python programming language (Python Software Foundation, https://www.python.org/). Estimation of the psychophysical thresholds and the subsequent experiments were conducted in a low-lit room with an enclosure around the monitor to ensure lighting remained consistent for all participants. Stimuli were presented on a 20-inch color monitor (resolution: 1920 x 1200 pixels; refresh rate 120 Hz) placed 50 cm in front of the participant and responses were recorded using the left and right arrow keys on a keyboard. A chin rest was used to stabilize the viewing distance and all participants made their responses with the right hand. All stimuli were presented in a pseudorandom order to ensure that identical stimuli were not presented consecutively.

Participants were instructed to indicate by key press as quickly and accurately as possible whether the target letter "H" was observed. The target could appear either as the small local letter or the large global letter (Fig 1), allowing the individual's implicit (i.e., uninstructed) preference for one level or the other to be assessed. The experiment began with one block of practice trials followed by the experimental trials. One third of the trials contained the target letter at the global level, one third of the trials contained the target letter at the local level, and one third of the trials served as catch trials with the target absent at both levels of processing. The stimuli were coded as Global (H made up of smaller distractor letters, either R or S), Local (H displayed as small letters forming a global S or R) and Neither (R or S made up of smaller distractor letters, either S or R respectively) for a total of 6 stimuli. The background of the

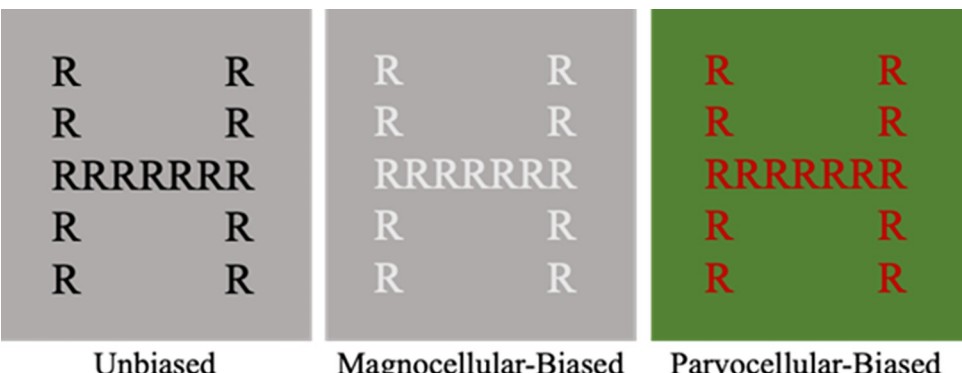

**Fig 1. The three types of stimuli used to bias processing.** This example shows incongruent stimuli in which the local letters (R) combine to form the global letter (H). The contrast and luminance properties of the magnocellular- and parvocellular-biased stimuli have been altered to make the stimuli more discernible to viewers.

hierarchical stimuli subtended 7.84˚ horizontally and 9.26˚ vertically, the global letters subtended a visual angle of 4.7˚ x 6.42˚ and the local letters subtended .73˚ x 1˚, respectively. The distance between the local letters was .2˚ visual angle. Initially these stimuli were presented as 'unbiased' (i.e., not biased towards the magnocellular or parvocellular pathways) hierarchical letter stimuli displayed as black letters on a grey screen. These 'unbiased' stimuli were converted into achromatic magnocellular-biased stimuli and red-green isoluminant parvocellular-biased stimuli using the psychophysical thresholds outlined previously, resulting in three conditions: unbiased, magnocellular-biased and parvocellular-biased. Each of the 6 stimuli were presented in each of the 3 conditions for a total of 18 stimuli.

In the practice block, each of the six stimuli were presented twice as unbiased stimuli for a total of 12 trials. For the experimental trials, the 18 stimuli were presented 10 times for a total of 180 trials. Each trial began with a fixation cross for 1000 ms, after which the stimuli were presented until a response was indicated. Reaction time (RT) and accuracy (ACC) were collected as dependent measures, and to account for both measurements an adjusted RT measure called inverse efficiency score (IES; [41, 42]) was calculated as: IES = RT/ACC and also used as a dependent measure. While conventional reaction time and accuracy measures provide statistics regarding the speed and error rates independent of one another, the addition of inverse efficiency score to the analysis provides a comprehensive summary of the findings by combining both measures together. IES takes into consideration differences in speed-accuracy trade-offs by adjusting reaction time performance for sacrifices in accuracy that might have been made in favor of speed. A mean reaction time achieved with a high accuracy will have a lower IES than the same reaction time achieved at the cost of more errors. To determine whether there were any perceptual differences between White and Asian participants attributed to relative differences in global versus local distribution of attention, 2 (Group: White, Asian) x 2 (Level: Global, Local) x 3 (Condition: Magnocellular, Parvocellular, Unbiased) ANOVAs were carried out on log-transformed reaction time and inverse efficiency scores. Reaction times and inverse efficiency scores were log-transformed to account for non-normality in the data. Since accuracy data was positively skewed and transformation of the data did not address concerns related to normality, a Poisson regression was used to analyze the number of errors made. All post-hoc pairwise comparisons were performed using Bonferroni correction and alpha = .05.

## Results

### Reaction time analysis

Results of the ANOVA for log-transformed reaction time showed significant main effects of level, $F(1,50) = 25.487$, $p < .001$, $\eta_p^2 = .338$, and condition, $F(2,100) = 401.206$, $p < .001$, $\eta_p^2 = .889$, and a significant level x condition interaction, $F(2,100) = 10.998$, $p < .001$, $\eta_p^2 = .180$. There was no significant main effect of group, $F(1,50) = 1.714$, $p = .196$, $\eta_p^2 = .033$, and no group x level x condition interaction, $F(2,100) = .197$, $p = .659$, $\eta_p^2 = .004$. Post-hoc t-tests on the level x condition interaction revealed faster identification of the target when it was presented globally compared to locally in the parvocellular condition ($p < .001$). Post-hoc tests also revealed that when the target was presented at the global level, it was identified faster in the unbiased condition compared to the magnocellular condition ($p < .001$), and in the parvocellular condition compared to the unbiased condition ($p < .001$). When the target was presented at the local level, it was identified faster in the unbiased condition compared to the parvocellular ($p < .001$) and magnocellular ($p < .001$) conditions. The target was also identified faster in the parvocellular compared to the magnocellular condition ($p < .001$).

## Accuracy analysis

A Poisson regression was run to predict the number of errors made based on whether the target was located in the global or local configuration (Level: Global/Local) and the condition type (Condition: Unbiased/Magnocellular/Parvocellular). Results of the Poisson regression using the Wald Chi-Square statistic showed significant main effects of level, $X^2(1) = 6.81$, $p = .009$, and condition, $X^2(2) = 6.7.46$, $p <, 001$, and significant interactions between level x condition, $X^2(2) = 13.56$, $p = .001$, and level x condition x group, $X^2(2) = 6.297$, $p = .043$. There was no significant main effect of group, $X^2(1) = 1.128$, $p < .001$ and no level x group interaction, $X^2(1) = .736$, $p = .391$.

Results of the level x condition interaction revealed the less errors were made when the target was presented at the global level compared to the local level in the magnocellular ($p = .015$) condition. No differences in accuracy were found in the parvocellular or unbiased conditions. For both global and local levels, the number or errors made was the highest in the magnocellular condition compared to the parvocellular (Global: $p = .001$; Local: $p < .001$) and unbiased (Global: $p = .001$; Local: $p < .001$) conditions. No accuracy differences were found between parvocellular and unbiased conditions.

Results of the three-way group x level x condition interaction revealed that for both Asian and White participants, when the stimuli was presented at the local level the number of errors made was found to be higher in the magnocellular condition compared to the parvocellular (Asian: $p = .03$; White: $p = .037$) and unbiased (Asian: $p = .002$; White: $p = .016$) conditions (Fig 2). No significant differences between conditions were found for either group when the target was presented at the global level.

## Adjusted reaction time analysis using inverse efficiency score (IES)

Results of the ANOVA on IES showed significant main effects of level, $F(1,50) = 27.665$, $p < .001$, $\eta_p^2 = .356$, and condition, $F(2,100) = 202.826$, $p < .001$, $\eta_p^2 = .802$, and significant

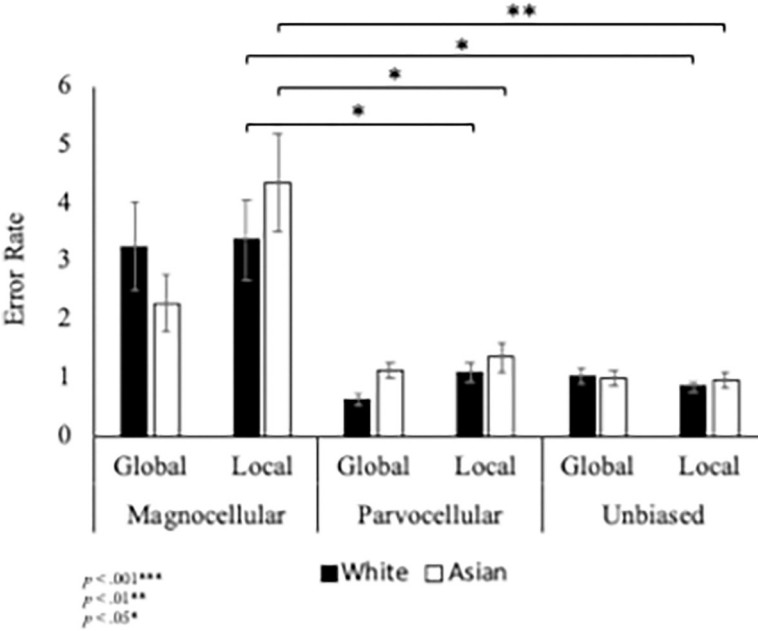

**Fig 2. Accuracy results.** More errors were made identifying the target in the magnocellular condition compared to the parvocellular and unbiased conditions when it was presented at the local level for both White and Asian individuals. Error bars show +/- 1 SEM.

interactions between level x group, $F(1,50) = 5.305$, $p = .025$, $\eta_p^2 = .096$, condition x level, $F(2,100) = 11.191$, $p < .001$, $\eta_p^2 = .183$, and group x condition x level, $F(2,100) = 2.977$, $p = .05$, $\eta_p^2 = .056$. There was no significant main effect of group, $F(1,50) = 1.700$, $p = .198$, $\eta_p^2 = .033$.

Results of the level x condition interaction revealed faster identification of the target when it was presented at the global level compared to the local level in both the magnocellular ($p = .001$) and parvocellular ($p < .001$) conditions. No differences were found in the unbiased condition. For both global and local levels the target was identified slower in the magnocellular condition compared to the parvocellular (Global: $p < .001$; Local: $p < .001$) and unbiased (Global: $p < .001$; Local: $p < .001$) conditions. A difference was also found between parvocellular and unbiased conditions at the local level ($p < .001$) but not at the global level. The level x group interaction revealed that within the Asian group, the target at the global level was processed faster than at the local level ($p < .001$). No differences were found between global and local processing for White participants.

Results of the three-way group x level x condition interaction revealed that in the parvocellular condition, when the target was presented at the global level, White participants responded more quickly than Asian participants ($p = .05$; Fig 3). In the unbiased condition, when the target was presented at the local level, White participants responded more quickly than Asian participants ($p = .045$). Results also showed faster processing for White participants when the target was presented at the global level compared to the local level ($p < .001$) in the parvocellular condition, while Asian participants showed faster processing at the global compared to the local levels in both magnocellular ($p < .001$) and parvocellular ($p < .001$) conditions. Finally, at the global level White participants showed slower processing of the target in magnocellular compared to parvocellular ($p < .001$) and unbiased ($p < .001$) conditions and for unbiased compared to parvocellular conditions ($p = .032$). At the local level, White participants also showed slower processing for magnocellular compared to parvocellular ($p < .001$) and unbiased ($p < .001$) conditions and for parvocellular compared to unbiased conditions ($p < .001$). At the global level, Asian participants showed slower processing in the magnocellular compared to parvocellular ($p < .001$) and unbiased ($p < .001$) conditions. No significant differences were found between parvocellular and unbiased conditions. At the local level, Asian

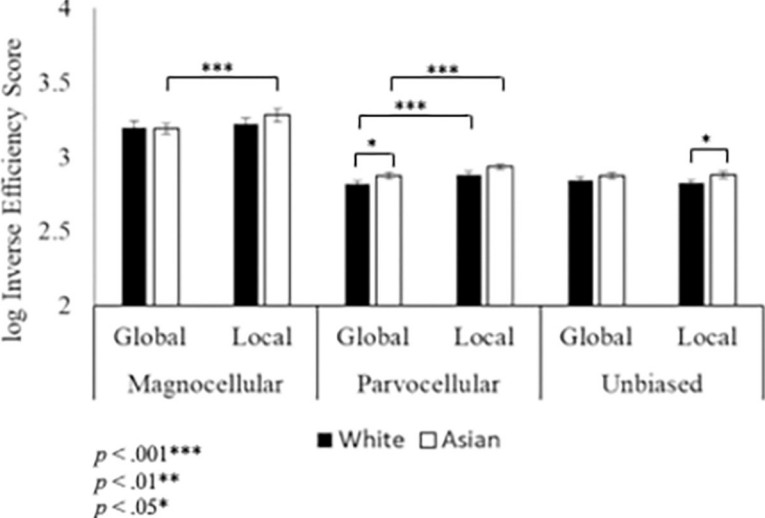

**Fig 3. Inverse efficiency score results.** Between groups, the target was identified faster at the global level compared to the local level in the parvocellular condition for White participants and in both the parvocellular and magnocellular conditions for Asian participants. Error bars show +/- 1 SEM.

participants showed slower processing in the magnocellular compared to parvocellular ($p <$ .001) and unbiased ($p <$ .001) conditions, as well as for parvocellular compared to unbiased conditions ($p <$ .001).

## Discussion

Previous studies suggest that East Asian individuals process scenes more globally than White individuals [15, 16]. Perceptual differences may be driven by culture-specific tuning towards visual spatial frequency information, as demonstrated in studies showing that White individuals preferentially process high spatial frequency information, while East Asian individuals preferentially process low spatial contextual information [17, 18]. Within the visual system, high spatial frequency information, required for local processing, is mainly conveyed via the parvocellular pathway and low spatial frequency information, required for global processing, is mainly conveyed via the magnocellular pathway. From this, we hypothesized that perceptual differences between White and Asian participants might be driven by biases towards one pathway or the other. The aim of this study was to test the potential mechanism underlying these differences by examining the relative contributions of the magnocellular and parvocellular pathways in mediating global and local visual processing in Asian and White participants.

Overall, our comparison of White and Asian participants revealed little evidence for cultural differences. In the unbiased condition, which would most closely represent the stimuli used in comparable previous studies, White participants performed similarly to those of Asian participants. Even more surprising, no global bias was found within each group. The lack of robust and consistent evidence is unexpected, given the original reports of statistically significant cultural differences. However, there are a number of reasons that could explain these differential findings.

First, stimulus presentation may have weakened our ability to see cultural differences. In previous research stimuli has been presented laterally rather than centrally [15]. While location of the stimuli was not found to influence global/local response times in their study, it is possible that faster reaction times for Asian compared to White participants were due to a broader allocation of attention in the Asian group rather than a global processing bias [5, 18]. In the current study, the stimuli were always presented centrally, so it may be the case that stimulus presentation was more advantageous for White participants, who have been shown to preferentially process information from foveal vision [17]. However, the number of errors made were low in both groups, suggesting that Asian participants were not that affected by the location of the stimulus.

A more plausible explanation is that differences in behavioral performance were masked by other factors such as attention. For example, if attention is viewed as a spotlight where stimuli falling within the beam of the spotlight are processed preferentially, then examining how the spotlight differs between groups may provide insight into why the two groups differ if there are characteristics that are unique to one group compared to the other, like size of the spotlight. While previous behavioral studies have shown that Asian individuals perform better on a task requiring global processing and White individuals perform better on a task requiring local processing [13], an fMRI study showed that equal levels of behavioral performance were achieved by allocating more sustained attentional effort for the local task in Asian individuals and the global task in White individuals [14]. In the latter case, this finding corresponds with activation in the frontal and parietal regions of the brain, which typically show greater activation for more demanding tasks and are thought to mediate cognitive control over working memory and attention [14]. As such, it is possible that White participants were recruiting more attentional resources to produce the same behavioral outcome as evident in the Asian

group. Future research using a combined fMRI/behavioral approach will help determine the extent to which attention influences global/local processing.

It is possible that individual biases may have influenced the extent to which global processing was observed. Previous research has shown that the degree of individual bias toward global information can vary based on stimulus parameters, such as the aspect ratio of local to global items [43, 44], the overall visual angle [45], or the amount of time participants have to view stimuli [46]. Additionally, older individuals [47], individuals induced into a state of negative affect [48], individuals from remote cultures [49] and musicians [50] all tend to show a local compared to global processing bias. Conversely, individuals from collectivist cultures [15] and individuals induced into a state of positive affect [51] tend to show a preference for global processing. Thus, a global bias can be influenced by participant characteristics and is not absolute. As such, culture is only one of many possible influencers of global processing, and future research involving larger sample sizes in each group and controlling for a wider breadth of participant characteristics will help to untangle the relationship between culture and global/local processing.

Finally, factors related to the two samples themselves may have limited the extent to which global processing was observed. While participants were asked to self-report their ethnicity and familiarity with the English language, other potentially important contributing information such as where participants were born, how long they had been residing in an English-speaking country, as well as factors related to socioeconomic status, possible influences from reading disorders, and familiarity of experience with digital technology were not collected, limiting the extent to which we can generalize these findings to the general population. While these results are promising and relevant, providing a deeper understanding of the potential mechanisms underlying a global processing advantage, considering the impact these factors may have on the current findings suggests that these findings should be considered preliminary and a foundation upon which future studies should be carried out.

Although no differences were observed in the unbiased condition, differences were found within the magnocellular and parvocellular conditions. Differences in reaction times and accuracy scores (measured by the number of errors made) emphasize the importance of considering both variables in the same measure, which was achieved using an inverse efficiency score. In this case, a mean reaction time achieved with a high accuracy will have a lower inverse efficiency score than the same reaction time achieved at the cost of more errors. A direct comparison between groups indicated lower inverse efficiency scores in White compared to Asian participants when processing the target at the global level, compared to the local level, in the parvocellular condition specifically. Individual group differences also showed that White participants processed the target with lower inverse efficiency scores at the global level compared to the local level in the parvocellular condition only, while no significant differences between global and local processing were observed in the magnocellular condition. Asian participants, however, processed the target with lower inverse efficiency scores at the global level compared to the local level in the magnocellular condition. In the parvocellular condition, Asian participants also showed lower inverse efficiency scores at the global level compared to the local level. Together, this suggests that global/local processing in White participants is influenced more by the parvocellular stream, and in Asian participants by the magnocellular stream (and to a lesser extent the parvocellular stream).

Based on our knowledge that the parvocellular pathway typically transmits high spatial frequency information and the magnocellular pathway is biased to convey low spatial frequency information, we suggest that the potential mechanism underlying global/local processing in White individuals relies more heavily on information from the parvocellular pathway, while the mechanism in Asian individuals relies more heavily on information from the

magnocellular pathway (and to a lesser extent the parvocellular pathway). This would explain why some research has found that White individuals show a local processing bias in global/local processing tasks (see [52]), and why White individuals show a less robust global processing bias compared to Asian individuals. While the current study biased parvocellular stimuli to convey low spatial frequency information so that global processing between the two pathways could be examined more directly, this strongly limited the extent to which we could interpret the results as the significant advantage shown by White compared to Asian individuals for processing global information when low spatial frequency information was conveyed through the parvocellular pathway may not have been as evident if high spatial frequencies were not filtered out of the stimuli. Consequently, these results can only be interpreted in the context of low spatial frequency stimuli, and future studies will be required to determine the extent to which White individuals process parvocellular stimuli when they are not filtered to isolate low spatial frequency information. Neuroimaging studies will also be an important contributor in verifying the extent to which these differences are observed in the brain.

A stronger influence of the magnocellular stimuli on global processing in the Asian group may also explain why these individuals show an early sensitivity to global information coding [16]. Magnocellular pathway neurons, critical for global processing, transmit information much faster than parvocellular pathway neurons from the lateral geniculate nucleus to the primary visual cortex. If visual processing in Asian individuals is more influenced by the magnocellular pathway than it is in White individuals, this suggests that a more robust global processing bias in Asian individuals may be driven by a stronger influence of the magnocellular pathway over the parvocellular pathway. This is also in line with previous research showing that Asian individuals rely more on low spatial, extra foveal vision.

Visual saliency between global and local features has also been suggested as a potential explanation for why White individuals process scenes less globally than Asian individuals. Previous work has found that White individuals were less efficient at detecting global compared to local feature changes, while Asian individuals performed equally well on both conditions, suggesting that the behavioral disadvantage of White individuals in the global task stemmed from differences in visual saliency between global and local features [16]. The rationale is that since visual processing of global features precedes the analysis of local information, an initial preference for global processing would conflict with local information, inhibiting the ability to detect local features [3, 53, 54]. Further research showed that participants identified local targets slower in the presence of a global shape, even when the global information was irrelevant [54]. From this, previous research suggests that when White individuals are required to detect changes in local information, the presence of global features is more disruptive for them than for Asian individuals, who seem to benefit from a top-down attention control to global features [16]. As such, the visual saliency induced by the global feature change did not seem to disturb processing in Asian individuals to the same extent as in White individuals. However, our results suggest that saliency cannot fully account for this difference in processing between the two groups since White individuals showed a global precedence in the parvocellular-biased condition using the same stimuli as was used in the unbiased and magnocellular-biased conditions. If visual saliency is the driving force behind a global processing difference, then Asian individuals should show the same global processing advantage over White individuals regardless of condition. It is more likely that the notion that White individuals process scenes less globally than Asian individuals is due to the way that visual information is conveyed by the two pathways. This is not to say that White individuals do not benefit from information transmitted via the magnocellular pathway, but rather suggests that the way in which visual information is transmitted via the two streams differs between groups.

Another potential factor that could influence magnocellular functioning is experience with digital technology. For example, several studies involving individuals with developmental dyslexia, in which magnocellular visual functioning is selectively deficient, have shown that reading difficulties associated with this visual deficit can be improved by playing action video games [55, 56]. This type of video game in particular involves a specific set of qualitative features in order to be successful in playing the game: extraordinary speed, an ability to take on a high degree of perceptual, cognitive and motor load to accurately maneuver through the game, keeping multiple action plans in memory and assessing each event presented in the game in the context of these action plans, and peripheral processing abilities [57]. As such, if individuals in the current study were avid video gamers, it is possible that differences in magnocellular functioning could be a result of familiarity with this type of gaming technology, a potential confound that should be considered in future studies.

It should also be noted that while the magnocellular and parvocellular stimuli were biased in a certain way, both types of cells may have still been responding. While magnocellular and parvocellular information conveyed from the lateral geniculate nucleus to the primary visual cortex does remain, to some extent, functionally segregated, once this information projects beyond V1 there is a considerable amount of mixing of magnocellular and parvocellular signals. From the primary visual cortex information is still conveyed via two largely functionally distinct streams, the dorsal and ventral streams, however each stream is composed of a mixture of magnocellular and parvocellular signals [58]. Although information in one stream is influenced by the other, lesion studies involving the magnocellular and parvocellular pathways provide evidence for a distinct relationship between the response properties of the magnocellular and parvocellular cells and the functions of the cortical regions along the dorsal and ventral streams. For example, lesions to the parvocellular pathway results in deficits in chromatic vision, texture perception, pattern perception, acuity and a loss in contrast sensitivity at low temporal and high spatial frequencies [59, 60]. Lesions to the magnocellular pathway have been found to cause deficits in flicker and motion perception [59]. Together, this demonstrates that while the dorsal and ventral streams receive input from both magnocellular and parvocellular cells, the effect these cells have on each of the two streams differs. As such, in the current study we do not assume that biasing the stimuli towards the magnocellular and parvocellular pathways only activates that individual pathway, but rather that the manipulation biases processing in one pathway over the other.

## Conclusion

We have demonstrated that White individuals can be biased to process scenes more globally than Asian individuals as long as low spatial frequency information is conveyed through the parvocellular pathway. Asian individuals also show a global precedence effect when low spatial frequency information is conveyed through the magnocellular pathway, and to a lesser extent through the parvocellular pathway. These findings suggest that a global processing advantage can be altered when stimuli is biased towards one of two subcortical pathways: the magnocellular or parvocellular pathway. That is, White individuals may depend more on ventrally-based information transmitted through the parvocellular pathway in global/local processing, while Asian individuals may depend more on dorsally-based information transmitted through the magnocellular pathway. However, since the parvocellular stream conveys high spatial frequency information useful for local processing when stimuli are not isoluminant, this may explain why research often finds that Asian individuals process scenes more globally than White individuals. Here we examined the more intricate mechanisms underlying global and local processing by looking individually at the two subcortical pathways that together form our

overall visual perception–the magnocellular and parvocellular pathways–demonstrating that when the parvocellular pathways is biased to only convey low-frequency information, this global advantage in Asian individuals can be reduced such that White individuals show a global advantage compared to Asian individuals. Of course, as humans we typically view visual stimuli in our environment as a combination of the inputs sent through both the magnocellular and parvocellular pathways, but in understanding the mechanism by which this global advantage is formed, it is important to consider the contribution of each pathway and how different types of spatial frequency information can alter visual output.

## Author Contributions

**Conceptualization:** Tiffany A. Carther-Krone, Jonathan J. Marotta.

**Formal analysis:** Tiffany A. Carther-Krone.

**Funding acquisition:** Jonathan J. Marotta.

**Investigation:** Tiffany A. Carther-Krone.

**Methodology:** Tiffany A. Carther-Krone.

**Supervision:** Jonathan J. Marotta.

**Writing – original draft:** Tiffany A. Carther-Krone.

**Writing – review & editing:** Tiffany A. Carther-Krone, Jonathan J. Marotta.

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
