## [Decision Letter · Decision Letter 0]

22 Dec 2021

PONE-D-21-29948The influence of magnocellular and parvocellular visual information on global processing in Caucasian and Asian populationsPLOS ONE

Dear Dr. Carther-Krone,

Thank you for submitting your manuscript to PLOS ONE. After careful consideration, we feel that it has merit but does not fully meet PLOS ONE’s publication criteria as it currently stands. Therefore, we invite you to submit a revised version of the manuscript that addresses the points raised during the review process.

As you could see from the reviewers' comments, your manuscript was considered relevant and satisfactory with respect to clarity but some information were missing in the rationale and methods. Specifically  the rationale could be better described and both reviewers highlighted that additional information is required in sample description, in the methods used and in the statistical procedures. About the statistics, I suggest  controlling for the distribution of the data (in particular the accuracy seems to be not normally distributed) and using non parametric model in the case of non normal distribution of the data.

I have also read the manuscript very carefully and I suggest adding in the rationale and in the discussion relevant information to the literature that does not support the theory with respect to the differences between magno and parvocellular segregation when dealing with behavioral methods. Alternative explanations of the results should also be given for completeness to the one provided. Moreover, I think that the significant results obtained are promising and relevant but, because of the small sample size recruited, I suggest to be more careful in the interpretation in terms of a characteristic of a general population (Asian vs Caucasian). Some additional limitations should be considered in the final discussion: are the sample matched for socio-economical status? and/or for the familiarity and experience of digital technology? Several studies highlight that magnocellular functions could improve significantly by the use of action video games, therefore it could be relevant to include this information in the discussion. In the light of the above mentioned limitations I suggest including in the discussion session that the results could be considered as preliminary.

We look forward to receiving your revised manuscript.

Kind regards,

Mariagrazia Benassi

Academic Editor

PLOS ONE

Journal Requirements:

2. Please note that according to our submission guidelines (http://journals.plos.org/plosone/s/submission-guidelines), outmoded terms and potentially stigmatizing labels should be changed to more current, acceptable terminology. For example: “Caucasian” should be changed to “white” or “of [Western] European descent” (as appropriate), including in the title and abstract. In addition, please change "female” or "male" to "woman” or "man" as appropriate, when used as a noun (see for instance https://apastyle.apa.org/style-grammar-guidelines/bias-free-language/gender).

This research was supported by a grant from the Natural Science and Engineering Research Council of Canada (NSERC) (Grant no. 04964-2018) held by J.J.M.

This research was supported by a grant from the Natural Science and Engineering Research Council of Canada (NSERC) (Grant no. 04964-2018) held by J.J.M.

Reviewers' comments:

Reviewer's Responses to Questions

**Comments to the Author**

1. Is the manuscript technically sound, and do the data support the conclusions?

Reviewer #1: Yes

Reviewer #2: Partly

2. Has the statistical analysis been performed appropriately and rigorously? 

Reviewer #1: Yes

Reviewer #2: I Don't Know

3. Have the authors made all data underlying the findings in their manuscript fully available?

Reviewer #1: Yes

Reviewer #2: Yes

4. Is the manuscript presented in an intelligible fashion and written in standard English?

Reviewer #1: Yes

Reviewer #2: Yes

5. Review Comments to the Author

Reviewer #1: This is an interesting study that addresses the culture differences in visual perception. Results show that Caucasians prefer to globally process low spatial frequency information that is presented via the parvocellular pathway, while Asians show a global processing when low spatial frequency information is presented via the magnocellular pathway. These findings suggest the association between global/local processing and subcortical pathways, advancing our understanding of the underlying mechanisms of culture differences in visual perception.

My concerns are listed below.

1) Please further explain how to distinguish magnocellular stimuli from parvocellular stimuli.

2) There are only 6 stimuli for each condition. The number of stimuli is not enough to achieve a reliable measure.

3) Why use IES? Please provide the detailed explanation for the use of such statistical method.

Reviewer #2: REVIEW

The study is interesting, well conducted and the manuscript is well written. Yet, there are some issues (and missing information) with the rationale behind the study and with the methodology which make it difficult to evaluate the actual soundness of the design and its publishable quality.

Methods:

The main question arising is: if comparing Caucasian and Asian participants who (probably) are readers of two (or more) different writing systems, why did the authors chose Navon’s letter task and not Navon’s geometric shapes task, which would have offered the same opportunities but not have suffered from any bias due to reading habits?

The absence of certain crucial conditions makes interpretation of the results very uncertain. For instance, a condition where P-function including low- and high-frequency information is missing (as also stated by the authors in the Discussion, suggesting that future research should provide this information). Thus, any hypothesis about what the results would have been with a “normal” P-condition are purely speculative.

Participants:

The information provided on participants’ characteristics is very poor and insufficient to evaluate their actual comparability. Where were the EA participants from? And the Caucasian ones? What language were they speaking and reading/writing? How long had they been in an English-speaking country? Were any dyslexic students among the participants? It is not sufficient to “assume” they were fluent in reading English: reading fluency should have been measured, compared with the Caucasians’ fluency and, if different, used as a covariate.

Rationale:

The rationale behind the study is not clear enough. The cultural and physiological differences seem to be addressed mixing up causes and effects and the reasoning is sometimes circular: how does culture affect physiology, and how does physiology affect culture? It is not clear whether the authors are proposing that physiological differences constrain functional patterns or that culture has influenced physiology through habits.

Minor points:

line 138-139: “we expected that in the parvocellular-biased condition Caucasians would process the target faster and more accurately at the global level than the local level”: the logic of this expectation should be explained more clearly

line 153: the sentence is redundant and circular (power of .8 and 80% power are the same information)

line 158: were expected to be highly fluent: you should state that no direct information is available

line 228: delete “are”

lines 370-372: what is the hypothesized role of sustained attention? I would guess that selective attention should be involved instead.

Lines 416-418: the absence of this condition with no filtering of low special frequency information is not an “optional addition” but a strong limitation of the study (making interpretation of results more difficult) and it should be described as such.

Lines 479-480: what do the authors mean by “depends on”?

Lines 486-488: the causal relationship (if any is hypothesized) should be made explicit – if no hypothesis is proposed, or if a parallelism without any causal relationship is hypothesized, it should be clearly stated.

6. PLOS authors have the option to publish the peer review history of their article (what does this mean?). If published, this will include your full peer review and any attached files.

Reviewer #1: No

Reviewer #2: No

---

## [Author Response · Author response to Decision Letter 0]

18 Apr 2022

Editor Recommendations:

1. About the statistics, I suggest controlling for the distribution of the data (in particular the accuracy seems to be not normally distributed) and using non-parametric model in the case of non-normal distribution of the data.

To control for the distribution of the data, reaction time and IES data were log-transformed and the analysis was re-run using the log transformed data. All results remained the same, and the statistics and figures have been updated to reflect the log-transformed data (p. 12-16).

Since accuracy in the form of error rate is considered count data and fails to fix the normality issue using a simple log transformation, a Poisson regression was used to analyze the count data (eg. The number of errors made). Since our data is positively skewed and contains a large proportion of zeros (eg. 0 errors made), the Poisson distribution is considered an appropriate statistical technique to address our normality issues. Most of the results remained the same, however, the level x group interaction was no longer significant after this analysis. This along with the corresponding pairwise comparisons have been removed, although this does not affect our results since they are based on the more comprehensive inverse efficiency score analysis. 

2. I suggest adding in the rationale and in the discussion relevant information to the literature that does not support the theory with respect to the differences between magno and parvocellular segregation when dealing with behavioral methods. Alternative explanations of the results should also be given for completeness to the one provided. Moreover, I think that the significant results obtained are promising and relevant but, because of the small sample size recruited, I suggest to be more careful in the interpretation in terms of a characteristic of a general population (Asian vs Caucasian). Some additional limitations should be considered in the final discussion: are the sample matched for socio-economical status? and/or for the familiarity and experience of digital technology? Several studies highlight that magnocellular functions could improve significantly by the use of action video games, therefore it could be relevant to include this information in the discussion. In the light of the above mentioned limitations I suggest including in the discussion session that the results could be considered as preliminary.

We agree that more demographic information would allow for better control over potential confounds in the analysis, and have noted that a lack of these variables, including information about socio-economic status and familiarity of experience with digital technology, along with factors outlined by reviewer 2 (where participants were from, how long they had been residing in an English speaking country, and more specific information regarding any disorders that could have impacted reading ability), is a limitation to the study in the discussion section, emphasizing the preliminary nature of these results (p. 18-19). 

Additionally, we have included a discussion about digital technology, with an emphasis on action video gaming, as an alternative explanation that could explain magnocellular functioning in the discussion (p. 22).

3. Please note that according to our submission guidelines, outmoded terms and potentially stigmatizing labels should be changed to more current, acceptable terminology. For example: “Caucasian” should be changed to “white” or “of [Western] European descent” (as appropriate), including in the title and abstract. In addition, please change "female” or "male" to "woman” or "man" as appropriate, when used as a noun (see for instance https://apastyle.apa.org/style-grammar-guidelines/bias-free-language/gender).

Caucasian has been changed to White individual (Asian has been changed to Asian individual for consistency).

4. Please provide additional details regarding participant consent. In the ethics statement in the Methods and online submission information, please ensure that you have specified what type you obtained (for instance, written or verbal, and if verbal, how it was documented and witnessed). If your study included minors, state whether you obtained consent from parents or guardians. If the need for consent was waived by the ethics committee, please include this information.

Updated to include “in writing” (p.8)

5. Please note that funding information should not appear in the Acknowledgments section or other areas of your manuscript. We will only publish funding information present in the Funding Statement section of the online submission form. Please remove any funding-related text from the manuscript and let us know how you would like to update your Funding Statement. Currently, your Funding Statement reads as follows: This research was supported by a grant from the Natural Science and Engineering Research Council of Canada (NSERC) (Grant no. 04964-2018) held by J.J.M. The funders had no role in study design, data collection and analysis, decision to publish, or preparation of the manuscript.

Funding information has been removed from the acknowledgments section. The funding statement as stands is correct.

Reviewer 1:

1. Please further explain how to distinguish magnocellular stimuli from parvocellular stimuli.

Further clarification of how magnocellular and parvocellular information were distinguished is provided at the end of the introduction (p.7).

2. There are only 6 stimuli for each condition. The number of stimuli is not enough to achieve a reliable measure.

The current study designed was modelled off of a prior study in which only 6 stimuli were used for each condition (magnocellular, parvocellular, unbiased) in Experiment 1, and 4 stimuli were used for each condition in Experiment 2 (Thomas et al., 2012). Even with their small sample size (5 healthy controls and 2 patients for Experiment 1 and 10 healthy participants for Experiment 2), a significant difference between conditions was observed. More recent studies have also employed a similar experimental set up (Guy et al. 2018: 4 stimuli: 2 congruent/2 incongruent; Primativo et al. 2020: 6 stimuli for each condition (parvocellular/unbiased); Baisa et al. 2021: 4 stimuli: 2 congruent/2 incongruent for each condition (small/medium/large).

References:

Guy, J., Mottron, L., Berthiaume, C., & Bertone, A. (2019). A developmental perspective of global and local visual perception in autism spectrum disorder. Journal of Autism and Developmental Disorders, 49(7), 2706-2720.

Primativo, S., Crutch, S., Pavisic, I., Yong, K., Rossetti, A., & Daini, R. (2020). Impaired mechanism of visual focal attention in posterior cortical atrophy. Neuropsychology, 34(7), 799.

Baisa, A., Mevorach, C., & Shalev, L. (2021). Hierarchical processing in ASD is driven by exaggerated salience effects, not local bias. Journal of Autism and Developmental Disorders, 51(2), 666-676.

3. Why use IES? Please provide the detailed explanation for the use of such a statistical method.

While conventional reaction time and accuracy measures provide statistics regarding the speed and error rates independent of one another, the addition of inverse efficiency score to the analysis provides a comprehensive summary of the findings by combining both measures together. IES takes into consideration differences in speed-accuracy trade-offs by adjusting reaction time performance for sacrifices in accuracy that might have been made in favor of speed. A mean reaction time achieved with a high accuracy will have a lower IES than the same reaction time achieved at the cost of more errors. 

This explanation has been updated in the methodology section on p. 11-12.

Reviewer 2:

1. Methods: The main question arising is: if comparing Caucasian and Asian participants who (probably) are readers of two (or more) different writing systems, why did the authors chose Navon’s letter task and not Navon’s geometric shapes task, which would have offered the same opportunities but not have suffered from any bias due to reading habits?

While it is entirely possible that Asian or Caucasian participants could have been proficient in two or more different writing systems, all participants were Introduction to Psychology students studying at an English language university (University of Manitoba) where applicants are required to demonstrate sufficient mastery of English before being accepted into a program in order to meet the demands of classroom instruction, written assignments and participations in oral discussions. Based on this rigorous requirement, all participants were expected to be highly fluent at identifying single English letters. By using the Navon’s letter task rather than the Navon’s geometric shapes task, this provided more consistency with prior studies, allowing for comparisons to be made between our study and studies that have examined a global advantage in Asian populations in the past (namely, McKone et al., 2010). 

2. The absence of certain crucial conditions makes interpretation of the results very uncertain. For instance, a condition where P-function including low- and high-frequency information is missing (as also stated by the authors in the Discussion, suggesting that future research should provide this information). Thus, any hypothesis about what the results would have been with a “normal” P-condition are purely speculative.

P.20 of the discussion has be re-written to acknowledge the absence of this condition with no filtering as a strong limitation to the study.

3. Participants: The information provided on participants’ characteristics is very poor and insufficient to evaluate their actual comparability. Where were the EA participants from? And the Caucasian ones? What language were they speaking and reading/writing? How long had they been in an English-speaking country? Were any dyslexic students among the participants? It is not sufficient to “assume” they were fluent in reading English: reading fluency should have been measured, compared with the Caucasians’ fluency and, if different, used as a covariate.

All participants were asked to indicate on their demographics form their ethnicity, as well as whether English was their first language and the languages they were fluent in. All Caucasians indicated English as their first language, while 11 Asians indicated English as their first language and 14 Asians indicated Chinese as their first language (Demographics updated on p. 8). All analyses presented in the manuscript were also carried out using a grouping variable in which Asians were split into two groups according to whether they spoke English as a first language or not, along with the Caucasian group, showing similar results to those outlined in the manuscript. Importantly, no significant differences were observed between the two Asian groups. However, the limitation to including these groups with much smaller sample sizes separately in the analysis resulted in decreased power and effect size. As such, the two groups were combined together into one Asian group to improve the power of our study.

While dyslexia was not queried specifically, as part of the demographics questionnaire, participants were asked to indicate whether they were diagnosed with any medical/psychological disorders. None were noted and thus no participants were excluded from the study.

We agree that more demographic information would allow for better control over potential confounds in the analysis, and have noted that a lack of these variables, including information about where participants were from, how long they had been residing in an English-speaking country, and more specific information regarding any disorders that could have impacted reading ability, is a limitation to the study in the discussion section (p. 18-19). 

4. Rationale: The rationale behind the study is not clear enough. The cultural and physiological differences seem to be addressed mixing up causes and effects and the reasoning is sometimes circular: how does culture affect physiology, and how does physiology affect culture? It is not clear whether the authors are proposing that physiological differences constrain functional patterns or that culture has influenced physiology through habits.

Here we are proposing that physiological differences influence behavior. While many cross-cultural studies point to habits and environment as driving forces behind physiological differences in global/local processing, here we aimed to look at more fundamental mechanisms underlying global and local processing as a potential explanation for differences in global/local processing between Asians and Caucasians. Since low-frequency spatial information is required for global processing, we hypothesized that a Caucasian preference for processing information through the parvocellular pathway (most notably involved in processing high-spatial frequency information important for local processing) is responsible for a lesser global processing response compared to Asians, and that this global processing ability could be strengthened by filtering out high-spatial frequency information. Since previous studies have found that Asians rely more on low contrast stimuli, involving the magnocellular pathway (important for global processing), by filtering out high-spatial frequency information from the parvocellular stimuli, this allowed us to compare global processing abilities overall and when stimuli are biased to one of the two subcortical pathways (magnocellular/parvocellular) as a way to examine if any differences in processing ability between the two subcortical pathways are responsible for the global processing advantage commonly found in the research. This clarification has been added to the end of the introduction on p. 7.

Minor points:

5. line 138-139: “we expected that in the parvocellular-biased condition Caucasians would process the target faster and more accurately at the global level than the local level”: the logic of this expectation should be explained more clearly

We have re-written this statement to clarify the rationale behind this expectation. (p.7)

6. line 153: the sentence is redundant and circular (power of .8 and 80% power are the same information)

“power of .8” has been removed to avoid redundancy

7. line 158: were expected to be highly fluent: you should state that no direct information is available

This information has been updated on p. 8.

8. line 228: delete “are” - deleted

9. lines 370-372: what is the hypothesized role of sustained attention? I would guess that selective attention should be involved instead.

In the context of the fMRI study to which the sustained attention refers to, the hypothesized role of sustained attention is thought to correspond with the activation found in the frontal and parietal regions of the brain, which typically show greater activation for more demanding tasks and are thought to mediate cognitive control over working memory and attention. In this case it is hypothesized that sustained attention is involved rather than selective attention because the stimuli in this study do not involve selecting and focusing on a particular aspect of the stimuli while simultaneously suppressing irrelevant or distracting information, but rather involve directly focusing on the specific stimuli. The difference in the amount of sustained attention allocated to stimuli between Asian and White individuals is the variable of interest in this case. This hypothesized role of sustained attention has been clarified on p. 17-18.

10. Lines 416-418: the absence of this condition with no filtering of low special frequency information is not an “optional addition” but a strong limitation of the study (making interpretation of results more difficult) and it should be described as such.

P.20 of the discussion has be re-written to acknowledge the absence of this condition with no filtering as a strong limitation to the study.

11. Lines 479-480: what do the authors mean by “depends on”?

In this case, “depends on” refers to an underlying factor influencing a global processing bias: the magnocellular or parvocellular pathway. To clarify, this sentence has been re-written to address this ambiguity (p.24).

12. Lines 486-488: the causal relationship (if any is hypothesized) should be made explicit – if no hypothesis is proposed, or if a parallelism without any causal relationship is hypothesized, it should be clearly stated.

Reframed the end of the conclusion to be more specific as to our intent behind the study (p.24).

---

## [Decision Letter · Decision Letter 1]

10 Jun 2022

The influence of magnocellular and parvocellular visual information on global processing in White and Asian populations

PONE-D-21-29948R1

Dear Dr. Carther-Krone,

We’re pleased to inform you that your manuscript has been judged scientifically suitable for publication and will be formally accepted for publication once it meets all outstanding technical requirements.

Kind regards,

Mariagrazia Benassi

Academic Editor

PLOS ONE

Additional Editor Comments (optional):

Reviewers' comments:

Reviewer's Responses to Questions

**Comments to the Author**

1. If the authors have adequately addressed your comments raised in a previous round of review and you feel that this manuscript is now acceptable for publication, you may indicate that here to bypass the “Comments to the Author” section, enter your conflict of interest statement in the “Confidential to Editor” section, and submit your "Accept" recommendation.

Reviewer #1: All comments have been addressed

2. Is the manuscript technically sound, and do the data support the conclusions?

Reviewer #1: Yes

3. Has the statistical analysis been performed appropriately and rigorously? 

Reviewer #1: Yes

4. Have the authors made all data underlying the findings in their manuscript fully available?

Reviewer #1: Yes

5. Is the manuscript presented in an intelligible fashion and written in standard English?

Reviewer #1: Yes

6. Review Comments to the Author

Reviewer #1: (No Response)

7. PLOS authors have the option to publish the peer review history of their article (what does this mean?). If published, this will include your full peer review and any attached files.

Reviewer #1: No

---

## [Editor Report · Acceptance letter]

16 Jun 2022

PONE-D-21-29948R1 

The influence of magnocellular and parvocellular visual information on global processing in White and Asian populations 

Dear Dr. Carther-Krone:

I'm pleased to inform you that your manuscript has been deemed suitable for publication in PLOS ONE. Congratulations! Your manuscript is now with our production department. 

Kind regards, 

on behalf of

Dr. Mariagrazia Benassi 

Academic Editor

PLOS ONE